# Postsurgical Analysis of Gait, Radiological, and Functional Outcomes in Children with Developmental Dysplasia of the Hip

**DOI:** 10.3390/s23073386

**Published:** 2023-03-23

**Authors:** Firdaus Aslam, Kamal Jamil, Ohnmar Htwe, Brenda Saria Yuliawiratman, Elango Natarajan, Irraivan Elamvazuthi, Amaramalar Selvi Naicker

**Affiliations:** 1Department of Orthopaedics and Traumatology, Faculty of Medicine, Universiti Kebangsaan Malaysia (UKM), Cheras, Kuala Lumpur 56000, Malaysia; 2IHT Rehabilitation Centre, Jalan Bioteknologi 1, Persiaran SILC, Kawasan Perindustrian SILC, lskandar Puteri 79200, Johor, Malaysia; 3Faculty of Engineering, Technology and Built Environment, UCSI University, Kuala Lumpur 56000, Malaysia; 4Department of Electrical & Electronic Engineering, Universiti Teknologi Petronas, Seri Iskandar 32610, Perak, Malaysia

**Keywords:** developmental dysplasia of the hip, gait analysis, inertial motion sensors, hip dislocation, radiological outcome

## Abstract

Background: Children undergoing DDH correction surgery may experience gait abnormalities following soft tissue releases and bony procedures. The purpose of this study was to compare the residual gait changes, radiological outcomes, and functional outcomes in children who underwent DDH surgery with those in healthy controls. Methods: Inertial motion sensors were used to record the gait of 14 children with DDH and 14 healthy children. Pelvic X-ray was performed to determine the Severin classification and the presence of femoral head osteonecrosis (Bucholz–Odgen classification). For functional evaluation, the Children’s Hospital Oakland Hip Evaluation Scale (CHOHES) was used. Results: There was no difference in spatial parameters between the two groups. In terms of temporal parameters, the DDH-affected limbs had a shorter stance phase (*p* < 0.001) and a longer swing phase (*p* < 0.001) than the control group. The kinematic study showed that the affected limb group had smaller hip adduction angle (*p* = 0.002) and increased internal rotation (*p* = 0.006) with reduced upward pelvic tilt (*p* = 0.020). Osteonecrosis was graded II, III, and IV in five, three, and one patients, respectively. Five patients had no AVN changes. The Severin classification was grade I, II, and III for six, three, and five patients, respectively. Most patients had good functional outcomes on the CHOHES, with a mean total score of 96.64 ± 5.719. Multivariate regression analysis revealed that weight, height, and femoral osteotomy were independent predictors for gait, radiological and functional outcome. Conclusion: Despite good functional scores overall, some children had poor radiological outcomes and gait abnormalities. Our results identified the risk factors for poor outcomes, and we recommend specified rehabilitative strategies for long-term management.

## 1. Introduction

Developmental dysplasia of the hip (DDH) is a spectrum of abnormal development that results in acetabular dysplasia, hip subluxation, and possible dislocation due to capsular laxity and other mechanical factors. It is known to be the most common orthopedic disorder in newborns, with an incidence of 1:100 for dysplasia and 1:1000 for hip dislocation [1]. However, the prevalence of DDH may vary depending on countries and continents. The incidence may vary from 1:1000 to 34:1000 [2,3,4,5]. An earlier study by Vascilcova et al. detected a strong effect of DDH on gait pattern among Saudi participants [6]. Most studies agree that the risk factors of DDH include being a woman, family history, firstborn status, and oligohydramnios. The management and treatment of DDH depends on the age of diagnosis. Early diagnosis before 6 months and management using a harness is the gold standard of treatment. In the event of delayed diagnosis or failure to use a harness, interventions such as closed or open reduction, with or without pelvic osteotomy, are performed. The procedure is performed to correct the abnormal morphology of the hips [7,8,9,10,11,12]. Despite efforts to improve the outcomes of DDH correction surgery, gait deficits are common following DDH surgery [13,14]. The study conducted by Lee et al. suggested that patients treated with unilateral DDH showed compromised, bilaterally different balance control strategies with an altered body center of mass and center of pressure during gait (frontal plane stance and sagittal plane swing) [15]. Gait analysis is typically performed in patients with hip joint surgeries to objectively evaluate walking function before and after surgery using temporospatial and kinematic parameters. However, there is little research on gait for DDH, as the outcome of surgery is mainly assessed by employing clinical examination, radiological outcomes, and functional parameters [1,16,17,18,19,20,21,22,23,24]. This can be seen in a study conducted by Jamil et al., who studied the functional outcomes of patients based on clinical examination and radiological outcomes [25]. There are also studies that used visual analysis to assess gait in participants with DDH, which requires physiotherapists who may assess differently from other physiotherapists involved [26].

In this study, we hypothesized that the surgically corrected DDH patients would show gait deviations after hip surgery. Information gathered from this study will help researchers understand and correlate gait deviations to the surgical procedure and the radiological and functional outcomes. Researchers will be able to improve their knowledge regarding the postoperative surgical outcomes of DDH and the impact on gait pattern. 

## 2. Materials and Methods

This study is a prospective case-control study conducted at a tertiary hospital for the duration of 18 months. A total of 14 DDH patients (unilateral/bilateral), who were surgically treated, were a minimum of 2 years postsurgery (irrespective of surgical method treatment), and aged between 8 and 18 years old were included in this study (Figure 1). The following were excluded from the study: patients with neurologic or cognitive impairments, unable to follow instructions for assessment; patients with infectious diseases, metabolic bone diseases, or other deformities in the lower limbs in addition to DDH, ambulating with any mechanical aid/orthosis; and patients with severe hip pain with limited mobility. For the control group, fourteen healthy individuals of similar age were recruited for this study.

For gait analysis, the patients were asked to walk at their own pace in a straight line on a 10 m walkway while wearing the Xsens Awinda inertial motion sensors (Xsens Technologies, Enschede, The Netherlands) (Figure 2). The data were then analyzed using MVN Analyze (Motioncloud, by Movella Incorporation (San Jose, CA, USA)) software where temporospatial and kinematics parameters were compared between the case and control groups. The central-edge angle (Severin classification) and the presence of osteonecrosis of the femoral head (Bucholz–Odgen classification) were recorded through the radiological evaluation of the pelvis. Children’s Hospital Oakland Hip Evaluation Scale (CHOHES) was used for clinical functional evaluation. The scale consists of three main domains: pain, functional, and physical examination. The maximum score is 100. 

Statistical analysis was performed using SPSS version 26 (IBM SPSS Statistics, IBM Corp., Armonk, NY, USA). Descriptive data re expressed as mean ± standard deviation (SD) unless otherwise stated. ANOVA was used to analyze categorical data, while the *t*-test was used to analyze numerical data with the means. Pearson correlation was performed to determine the association between the risk factors and the outcome measures. A difference was considered statistically significant at *p* ≤ 0.05. 

## 3. Results

### 3.1. Patient Demographics

The characteristics of the participants are described in Table 1. No significant differences between the demographic characteristics of the two groups were observed.

### 3.2. Gait Outcome

In terms of spatial parameters such as speed, cadence, step width, step length, and stride length, no difference between the DDH group and the control group was observed. Figure 3 shows the outcomes of the range of motion analysis among the children with DDH. Kinematic studies on the range of motion of the pelvis, hip, knee, and ankle demonstrated significant differences between the affected limbs and those of the control group: the minimum hip abduction angle (adduction) had a deviation mean value of −3.25 ± 3.67 for the affected group compared with −6.18 ± 2.28 (*p* = 0.002) for the control group; and the maximum hip rotation (internal rotation) had a deviation mean value of 4.96 ± 5.48 for the affected group compared with 0.43 ± 3.89 (*p* = 0.006) for the control group. The DDH-affected limb appeared to have significantly less hip adduction when walking compared with the limbs of the control group, and even with the unaffected limb (0.045). In terms of the compensatory effect of the DDH-affected limb on the unaffected limb, no significant kinematic parameters could be observed. In terms of knee and ankle flexion and extension, no significant difference could be seen for the deviation mean between the affected and control groups. Regarding the pelvic motion parameters, it was seen that the affected limb group had less upward hiking compared with the limbs of the control group in the coronal plane (*p* = 0.020).

For temporal parameters, there was a significant increase in swing phase duration, swing phase per gait cycle, single support phase duration, single support phase per gait cycle, midstance duration, and midstance per gait cycle in the DDH group compared with in the control (Table 2). Simultaneously, significant reductions in stance phase per gait cycle, double support phase duration, double support phase per gait cycle, loading response phase duration, loading response phase per gait cycle, and preswing duration were noted in the affected group compared with the control group. The compensatory effect of DDH on the unaffected limb was not apparent in any of the parameters.

### 3.3. Radiological Outcomes

With regard to the radiological outcomes of the 14 patients, 5 (No AVN), 5 (grade II), 3 (grade III), and 1 (grade IV) patients were classified according to the Bucholz and Olden classification. In contrast, six (grade I), three (grade II), and five (grade III) patients were classified according to the Severin classification.

Measurements of intra- and interobserver variability obtained an intraclass correlation coefficient (ICC) value of 0.96 and 0.93, respectively, with a 95% confidence interval. This indicated excellent reliability because the ICC values for both observers are greater than 0.75.

### 3.4. Functional Outcomes

Utilizing the CHOHES functional assessment tool, most of the patients exhibited good functional outcomes with a mean of 96.64 ± 5.719 in total score. The mean scores for each of the domains, which consist of pain, function, and physical examination, were 39.29, 30.57, and 26.79, respectively. For the range of motion outcomes obtained from the physical examination among the 14 children with DDH, the results were as follows: (a) hip internal rotation, 0 (<16°), 1 (16–29°), 1 (30–39°), and 12 (>39°); (b) hip external rotation 0 (<16°), 1 (16–29°), 2 (30–39°), and 11 (>39°); (c) hip flexion 0 (<90°), 2 (90–100°), 5 (101–114°), and 7 (>114°); and (d) hip abduction 0 (<20°), 0 (21–29°), 0 (30–39°), and 14 (>39°), respectively.

### 3.5. Risk Factors Associations with Outcome in Children with DDH

The association of each patient’s characteristics with the measured outcome is depicted in Table 3 to identify the risk factors of adverse outcomes following DDH. Changes in the minimum and maximum pelvis tilt, gait cycle duration, CHOHES physical examination score, and CHOHES total score were found to be associated with variations in terms of the patient’s sex, type of surgery, age during surgery, weight, and height. Additionally, changes in step duration were found to be associated with age during surgery, weight, and height. In terms of CHOHES pain score, it was found to be associated with age during surgery and weight of the patient but not their height. Moreover, the CHOHES function score and both radiological outcomes were found to be associated with the type of surgery, age during surgery, and height and weight of the patient. Finally, the side of the affected limb showed no association with any of the outcomes measured.

The multiple linear regression analysis to control the confounding effects of the variables is shown in Table 4. In terms of gait outcome, weight had an independent correlation with the minimum pelvis backwards tilt, the maximum pelvis forward tilt, and the gait cycle duration. In detail, an increase in weight resulted in reductions in the minimum and maximum pelvis tilts, subsequently causing a delay in the gait cycle duration. In addition, those who had femoral osteotomy and were older at surgery tended to walk faster than those who had other types of surgery (pelvic osteotomy, open reduction, or combined surgery), as evidenced by the shorter gait cycle duration. Moreover, children who were older during their surgery were also found to have shorter stance phase duration in their affected limb. For the Severin classification, children who underwent femoral osteotomy and combination surgery were found to have lower odds of having severe deformity of the hip joint. For the CHOHES score, only an increase in age during surgery and weight was found to reduce the odds of having high scores in the CHOHES function and physical examination domain. 

## 4. Discussion

The results of the current study demonstrated the existence of gait deviations in the surgically corrected DDH patients group compared with the control group. Furthermore, the comparable demographic characteristics between the case and control groups justified the use of this population as a baseline for healthy gait. Although comparing the postoperative gait of DDH children with healthy children might be less than ideal, a comparison of the pre- and postoperative gait in the same patient was not possible as the patients had their surgery before, or at, walking age [27]. Nevertheless, a comparison of the postoperative gait with the unaffected limb within the DDH children group could be performed to look for any compensatory mechanisms of the contralateral limb (1).

As for the spatial parameters in the gait analysis (speed, cadence, step width, step length, and stride length), no significant difference was seen between the DDH and the control groups. A similar outcome was also reported by Ömeroğlu et al., where the mean velocity and step-length values of the soft tissue release surgery-affected and unaffected sides were similar to those of the healthy control group [27]. In another study among 40 patients who underwent total hip arthroplasty (THA), it was also observed that the mean temporospatial values recorded in terms of speed, cadence, step width, step length, and double-step length were similar between the operated sides and nonoperated sides [28].

In terms of the kinematic parameters, the evaluation of the range of motion of the pelvis, hip, knee, and ankle in the coronal, sagittal, and axial plane revealed an increase in hip internal rotation during walking in the affected hip when compared with those of the control group. This could be due to DDH patients’ abnormal proximal femoral morphology in the first place. DDH patients have an abnormal rotational profile at the proximal femur, which may cause an increase in internal rotation in the affected hip in addition to a dysplastic acetabulum [29,30]. By using EOS imaging technology, which is a low-dose radiation and weight-bearing X-ray technology, Passmore et al. described this correlation when they demonstrated an increase in internal hip rotation during the stance phase in children’s gait following the increase in femoral neck anteversion [30].

When compared with the control group, the affected hip showed reduced adduction during walking. This could be attributed to the weakness of the hip adductor of the affected hip, because adductor tenotomy was performed during surgery to help reduce the hip and increase the safe zone of Ramsey for postoperative application of the hip spica. Interventions to strengthen hip adductor muscle strength and flexibility, particularly static stretching, have demonstrated restoration of hip adduction in 40 patients with limited hip adduction [31]. Thus, interventions for hip adductor muscle strengthening should be prescribed for a patient with DDH in order to restore the normal range of hip adduction motion.

A weak hip abductor, which is common in hip pathology, may also cause an increase in hip internal rotation. Weakness in the hip abductor could explain the reduction in pelvic upward tilt in the coronal plane observed in this study. Compared with the normal control group, the DDH-affected hip showed a statistically significant reduction in upward pelvic tilt. Similar findings were described in DDH patients in their adulthood when compared with the control population following triple innominate osteotomy or total hip arthroplasty procedures [32,33]. Thus, strategies for the strengthening of the hip abductor muscle should be prescribed in this population to provide a better prognosis in the long run.

Regarding gait temporal parameters, the DDH-affected hip had a shorter stance duration and percentage, as well as a longer swing phase duration and percentage, when compared with the limbs of the normal group. This finding could be attributed to patients offloading their pathological hip earlier in order to avoid causing discomfort to the affected hip (1). Chang et al. obtained a similar result, with a significantly shorter terminal double limb stance (DLS) in the affected limb after Pemberton osteotomy. Similarly, Nie et al. reported shorter DLS and single limb stance (SLS) among DDH patients who underwent unilateral THA surgery compared with the normal healthy individual, although the differences were not significant between the two groups [21]. When compared with the control group, DDH-affected limbs showed an increased midstance duration (s) and midstance per gait cycle (%). According to Wadsworth hip biomechanics, maximum hip abduction occurs during the midstance phase. If the hip abductor muscles are weak, this could explain the increase in midstance duration and midstance per gait cycle [34]. Single support phase duration (s) and single support per gait cycle (%) were also longer in the DDH-affected limb than in the control group, which could have been due to the affected hip’s weak hip abduction requiring more time to offload the affected hip from the ground.

The strength of the current study relies on the multivariate regression analysis of risk factors influencing the outcome of DDH correction. This allows the identification of independent predictors of the outcome of DDH correction after adjusting for other confounding factors. In terms of gait variables, it was revealed that increasing weight independently predicted the increase in downward and backward pelvis tilt. Furthermore, increasing weight was also found to be an independent predictor of an increase in gait cycle duration (s), while greater height and femoral osteotomy were found to reduce the gait cycle duration. Finally, increasing age at surgery resulted in a decrease in the stance phase duration (s). The interactions between all the aforementioned risk factors explained the consequence of the late intervention of DDH.

After adjusting for other confounding factors that may influence the Severin classification outcome (such as the age of surgery, and height and weight of the child), femoral or combination (femoral and pelvic) osteotomy and open reduction surgery were found to be independent predictors of lower odds of having severe deformity of the hip joint. However, this finding could have been influenced by the single sample available for each surgery type. The relationship between the radiological outcome and the type of surgery varies in the literature. Better radiological outcomes have been associated with open reduction surgery alone without the additional bony procedure, because children requiring more complex surgery may already have poor baseline radiological outcomes compared with those who underwent open reduction alone [35]. Nevertheless, good radiographic outcomes have been reported among patients undergoing open reduction in combination with pelvic osteotomy compared with open reduction alone [36]. Additionally, no difference between the radiographic outcome among children with DDH who underwent open reduction alone and those with additional bony procedures was previously reported [25].

Moreover, increased age at surgery and weight were found to be associated with a reduced likelihood of having a high CHOHES score. In this study, when age was considered as a categorical value (being younger or older than four years during surgery), there was a significant difference (*p* < 0.05): the mean CHOHES score was recorded as 95.7 (84–100) for patients younger than four years during surgery compared with 92.5 (82–100) for patients older than four years during surgery. Similarly, numerous studies have established age as a predictor of poor clinical outcomes [25,37,38,39]. Castaneda et al., in particular, reported a trend toward lower CHOHES scores with increasing age [36]. In addition, the current study showed increased weight led to a higher odds of lower CHOHES function (OR = 3.55) and physical examination (OR = 3.23) scores.

In summary, although our patients demonstrated good functional outcomes, there were significant gait abnormalities that may affect these children in the long run. However, there are some limitations in this study. First, the sample size is relatively small; hence, the actual amplitude of gait deviation in postsurgically treated DDH patients may not have been adequately collated. Moreover, this is a retrospective study analyzing the outcomes of various surgeries; thus, the outcomes are fairly heterogenous. Further research conducted in a larger population with standardization of the type of surgery performed may produce more accurate results. Ideally, a follow-up gait analysis can also be performed following rehabilitative interventions. Despite all these limitations, our findings suggest that gait analysis utilizing inertial motion sensors is able to detect changes that may not be discovered in routine physical examinations.

## 5. Conclusions

The analysis of gait, functional, and radiological outcomes among children with DDH in the current study allowed for the identification of the risk factors associated with each outcome. The most important risk factors associated with most of the outcomes were the type of surgery, age during surgery, height, and weight. Surgically corrected DDH patients showed significant gait deviations, particularly in temporal parameters; therefore, our hypothesis as accepted. Consequently, information from the study can be utilized for rehabilitation and to aid researchers in devising strategies to improve prognosis and gait among children with DDH. Interventions aimed at strengthening the affected muscles that may affect the gait in DDH patients can be derived from the results presented. Gait analysis utilizing an inertial motion sensor is recommended to identify subtle gait changes in DDH patients.

## Figures and Tables

**Figure 1 sensors-23-03386-f001:**
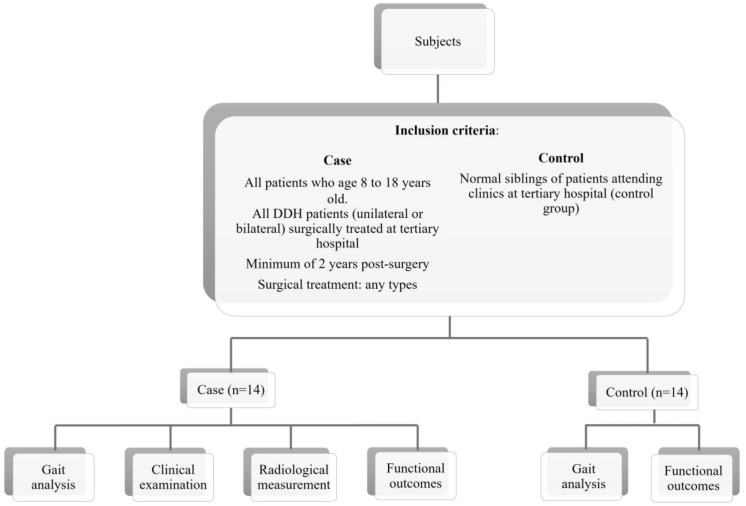
Patients’ recruitment pathway and methodology.

**Figure 2 sensors-23-03386-f002:**
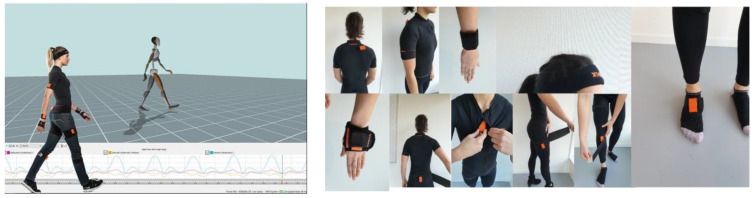
Xsens Awinda inertial motion sensors and its location.

**Figure 3 sensors-23-03386-f003:**
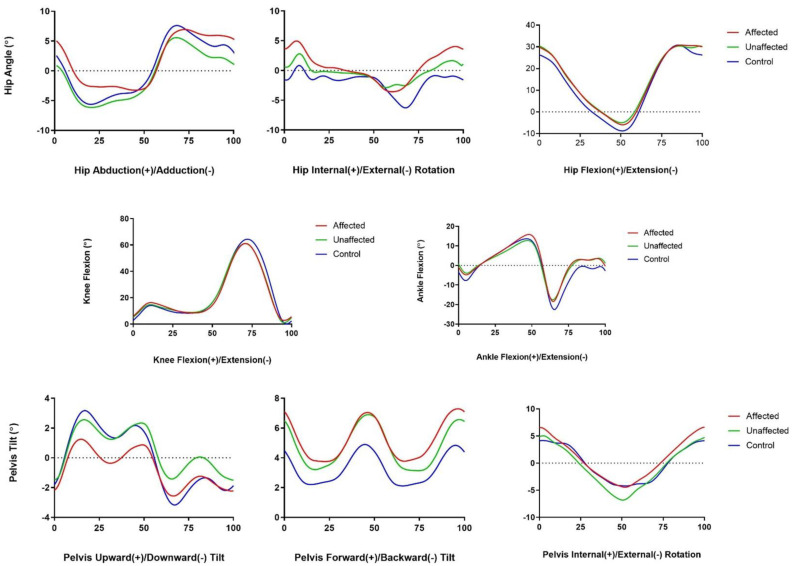
Mean range of motion of the hip, pelvis, and knee throughout the gait cycle among the participants.

**Table 1 sensors-23-03386-t001:** Characteristics of the participants.

	Patients with DDH (n = 14)	Controls (n = 14)	*p*-Value
Age at follow up	12.00 ± 2.83	12.79 ± 2.16	0.416
8 years old and younger	2 (14.3%)	0 (0%)	
9–11 years old	4 (28.6%)	4 (28.6%)	
12–14 years old	6 (42.9%)	7 (50%)	
15 years old and older	2 (14.3%)	3 (21.4%)	
Height	1.43 ± 0.11	1.46 ± 0.07	0.440
Weight	51.02 ± 18.52	40.86 ± 11.28	0.091
Sex			
Male	3 (21.4%)	3 (21.4%)	
Female	11 (78.6%)	11 (78.6%)	
Type of surgery			
Open reduction	7 (50%)		
Femoral osteotomy	1 (7.1%)		
Pelvic osteotomy	5 (35.7%)		
Combined osteotomy	1 (7.1%)		
Age during surgery	33.36 ± 27.89		
24 months old and younger	6 (42.8%)		
24–36 months old	4 (28.6%)		
37–48 months old	2 (14.3%)		
49 months old and older	2 (14.3%)		
Affected hip side			
Right	4 (28.6%)		
Left	10 (64.3%)		
Bilateral	1 (7.1%)		

Note: categorical values are expressed as frequency (percentage), while continuous values are expressed as mean ± standard deviation.

**Table 2 sensors-23-03386-t002:** Deviation in terms of temporal parameters between affected limbs of patients with DDH compared with control and unaffected limbs.

Measurement	Mean Value among Children with DDH		Mean Value among Control Children	Affected vs. Control	Affected vs. Unaffected
	Affected	Unaffected		*p*-Value	*p*-Value
Swing phase duration (s)	0.49 ± 0.06	0.49 ± 0.07	0.43 ± 0.02	**<0.001 ****	0.877
Swing phase per gait cycle (%)	44.62 ± 4.80	43.88 ± 4.86	40.49 ± 1.45	**<0.001 ****	0.687
Stance phase per gait cycle (%)	51.97 ± 14.98	56.11 ± 4.93	59.48 ± 1.46	**0.011 ***	0.351
Single support phase duration (s)	0.49 ± 0.07	0.49 ± 0.06	0.43 ± 0.02	**<0.001 ****	0.785
Single support per gait cycle (%)	44.77 ± 5.20	43.67 ± 4.31	40.48 ± 1.44	**<0.001 ****	0.550
Double support phase duration (s)	0.07 ± 0.04	0.08 ± 0.05	0.10 ± 0.02	**0.001 ***	0.608
Double support per gait cycle (%)	6.20 ± 3.68	6.89 ± 3.93	9.50 ± 1.50	**<0.001 ****	0.636
Loading response duration (s)	0.07 ± 0.04	0.08 ± 0.05	0.10 ± 0.02	**0.001 ***	0.608
Loading response per gait cycle (%)	6.20 ± 3.68	6.89 ± 3.93	9.59 ± 1.63	**<0.001 ****	0.636
Midstance duration (s)	0.24 ± 0.06	0.23 ± 0.04	0.20 ± 0.02	**0.002 ***	0.579
Midstance per gait cycle (%)	22.08 ± 5.47	21.02 ± 3.35	18.80 ± 2.65	**0.011 ***	0.550
Preswing duration (s)	0.07 ± 0.05	0.08 ± 0.04	0.10 ± 0.02	**0.002 ***	0.756

Note: values expressed as mean ± SD. ** and *: significance difference (** *p* < 0.01 and * *p* < 0.05, respectively) gait deviation through unpaired *t*-test.

**Table 3 sensors-23-03386-t003:** Association between risk factors and measured outcome.

Outcomes	Gender	Type of Surgery	Age during Surgery	Height	Weight
**Gait Outcome**					
Min Pelvic Tilt (Backward)	**0.022 ***	**<0.001 ****	**<0.001 ****	**<0.001 ****	**<0.001 ****
Max Pelvic Tilt (Forward)	**0.032 ***	**0.001 ***	**<0.001 ****	**<0.001 ****	**<0.001 ****
Gait Cycle Duration	**0.014 ***	**0.004 ***	**<0.001 ****	**<0.001 ****	**<0.001 ****
Step Duration	0.100	0.179	**0.049 ***	**0.014 ***	**0.031 ***
Stance Phase Duration	0.270	0.099	**0.021 ***	0.100	0.050
**Radiological Outcome**					
Bucholz and Olden Classification	0.236	**<0.001 ****	**<0.001 ****	**0.020 ***	**<0.001 ****
Severin Classification	0.712	**0.003 ***	**0.001 ***	**0.039 ***	**0.001 ***
**Functional Outcome**					
CHOHES Pain	0.443	0.146	**0.009 ***	0.545	**0.009 ***
CHOHES Function	0.372	**0.003 ***	**0.001 ***	**0.010 ***	**0.001 ***
CHOHES Physical Examination	**0.037 ***	**<0.001 ****	**<0.001 ****	**0.001 ***	**<0.001 ****
Total CHOHES Score	**0.007 ***	**<0.001 ****	**<0.001 ****	**<0.001 ****	**<0.001 ****

Note: values expressed as Pearson’s correlation coefficient/X^2^, *p*-value. ** and *: significance difference (** *p* < 0.01 and * *p* < 0.05, respectively) of respective variables through unpaired *t*-test.

**Table 4 sensors-23-03386-t004:** Multivariate linear regression analysis between risk factors and measured outcomes.

Outcomes	Age during Surgery OR (CI; *p*-Value)	Height OR (CI; *p*-Value)	Weight OR (CI; *p*-Value)	Femoral Osteotomy OR (CI; *p*-Value)	Combined Osteotomy OR (CI; *p*-Value)
Min Pelvic Tilt (Back)			**−5.37 (−1.85–−1.54; 0.023 *)**		
Max Pelvic Tilt (Forward)			**−5.25 (−2.02–−0.25; 0.014 *)**		
Gait Cycle Duration		**−2.00 (−2.97–−0.86; 0.001 *)**	**7.43 (0.03–0.06; <0.001 *)**	**−0.95 (−0.29–−0.13; <0.001 *)**	
Stance Phase Duration	**0.42 (0.00–0.00; 0.028 *)**				
Severin Classification				**0.68 (0.54–2.12; 0.002 *)**	**0.49 (0.27–2.18; 0.015 *)**
CHOHES Function	**−1.06 (−0.13–−0.03; 0.004 *)**		**−3.55 (−0.81–0.01; 0.045 *)**		
CHOHES PE	**−0.75 (−0.07–−0.00; 0.028 *)**		**−3.23 (−0.43–−0.03; 0.029 *)**		

Note: * indicates relationship significance (* *p* < 0.05) with multivariate linear regression.

## Data Availability

The data presented in this study are available on request from the corresponding author. The data are not publicly available due to privacy and ethical reasons.

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
