# Peer review of "Postsurgical Analysis of Gait, Radiological, and Functional Outcomes in Children with Developmental Dysplasia of the Hip"

_sensors, 2023, doi:10.3390/s23073386_

Round 1

Reviewer 1 Report

The paper's topic is comparing results regarding gait changes and radiological and functional outcomes in 14 children who underwent developmental dysplasia of hip surgery to healthy controls (also 14). This was done using inertial motion sensors and statistics.

The substantial area is the topic. Monitoring children after the surgery seems to be crucial to give them proper rehabilitation.

Weaknesses: small sample, the article could be more straightforward for people not related to this area — the method of measurement needed to be explained (location of the sensors).

My comments:

In abstract:

33- "Our results identified the risk factors for poor outcome and recommended specified rehabilitative strategies for  long-term management"  - Which strategies do you recommend in the paper?

93 - spatial parameters should be listed here, not in line 179

94 - shows

159- (for, in case of) other types of surgery?

194  - EOS abbreviation is not explained

213 – Is the strategy for strengthening the hip abductor used in rehabilitation now?

277 – "The analysis of gait, functional and radiological outcome among children with DDH in the current study allows for the identification of the risk factors associated with each

 outcome." The most important risk factors are …..

(I think that it should be clearly stated here)

In line 49, you hypothesized that the surgically corrected DDH patients would show gait deviations after the postoperative hip surgery. Could you clearly and briefly write in the conclusions what are the most critical deviations that can confirm your hypothesis

Author Response

We appreciate the time and efforts by the editor and reviewers in reviewing this manuscript. We have addressed all the issues indicated in the review report, and hopefully the revised version can meet the journal publication requirements

Reviewer 1 comments:

he paper's topic is comparing results regarding gait changes and radiological and functional outcomes in 14 children who underwent developmental dysplasia of hip surgery to healthy controls (also 14). This was done using inertial motion sensors and statistics.

The substantial area is the topic. Monitoring children after the surgery seems to be crucial to give them proper rehabilitation.

Weaknesses: small sample, the article could be more straightforward for people not related to this area — the method of measurement needed to be explained (location of the sensors).

My comments:

In abstract:

33- "Our results identified the risk factors for poor outcome and recommended specified rehabilitative strategies for long-term management"  - Which strategies do you recommend in the paper?

Thank you for the reviewer’s concern.  The authors believed that recommendations have been made at line 204-206 and 213-214. As authors found few risks factor associated with poor outcomes, authors believed that strengthening of the hip abductor muscle is one of few strategies that can be done for good outcome in long term.

Besides that, the authors also believed that a combination of open surgery and pelvic osteotomy provide a good outcome compared to open surgery alone.

93 - spatial parameters should be listed here, not in line 179

Thank you for the comment. Corrections have been made at the line of 93.

94 - shows

Thank you for the comment. This is a typographical error. Correction has been made accordingly.

159- (for, in case of) other types of surgery?

Thank you for the comment.  Changes have been made accordingly at line 161.

194  - EOS abbreviation is not explained

Thank you for the comment. Authors believed EOS imaging technology is imaging technology that developed by EOS imaging company which is why EOS abbreviation is not explained.

213 – Is the strategy for strengthening the hip abductor used in rehabilitation now?

Thank you for the comment. The author believed that the current strategy used in rehabilitation regards to study sample age (8-18 years old) is exercise, weight loss and hippotherapy which does not directly aim to increase the hip abductor strength.

277 – "The analysis of gait, functional and radiological outcome among children with DDH in the current study allows for the identification of the risk factors associated with each outcome." The most important risk factors are …..

(I think that it should be clearly stated here)

Thank you for the comment. Changes have been made accordingly in line 279-280.

In line 49, you hypothesized that the surgically corrected DDH patients would show gait deviations after the postoperative hip surgery. Could you clearly and briefly write in the conclusions what are the most critical deviations that can confirm your hypothesis

Thank you for the comment. Changes have been made accordingly in line 280-282.

Reviewer 2 Report

Overall: A full journal paper should cite approximately 40 relative references and minimal word count of 4000 words (please double check). 

Introduction: 

1.     The authors included background in the introduction section, but there is a lack of literature review in this study. The latest reference was cited in the introduction is 2020. There are many current studies focusing on DDH if you type the key words in Google scholar. The authors should essentially explain what the previous studies have done, and what their limitations were, and what the main contribution is in this study. Hence, the introduction section should be significantly improved, and should include more current references.

I have a big concern about the study design and results.

Methods:

1.     How did you measure the speed, cadence, step width, step length and stride length, and how did you compare them?

2.     Figure 1 is redundant. It repeated the contents in the first two paragraphs.

3.     Figure 2 is not clear. Xsens Awinda should not have wires connecting sensors, but why the subject has wires secured?

4.     Given that the subjects have been recorded, why used the picture of Xsens from website, and not use the pictures of subject recruited in this study during their task performance?

5.     Stats analysis is not clear. T-test to analyze the mean values seems not correct. If you have two groups and each group has 2 subjects, the body weights are 120kg and 60kg in the first group, and 90kg and 90kg in the second group. The mean values are same, but the difference should be significantly different. 

Results:

1.     For the stats analysis p=0.09, how did you get this result? There is ~10kg difference between these two groups. It is a big difference.

2.     For the figure 3, there are 3 curves in each plot. Are the 3 curves obtained from one subject or the average curves from all 14 subjects?

3.     The description is not accurate for the statement of minimum hip abduction. For example, the minimum hip abduction for the red line is at 50 degrees. I don’t think there is a significant difference.

4.     If the authors put knee and ankle results in the paper, you should describe them. But I did not see the description in the result section.

5.     I have a big concern about the subjects recruited in both groups. To compare the affected results and the control results for all the joints, there should not be a significant difference between the unaffected results and the control results between two groups. Otherwise, it will be difficult to identify the factors or obtain the conclusions for the comparison between the affected and control results. But from figure3, I can see some significant differences between the unaffected results and the control results. Moreover, the difference of anthropometric data between subjects may also lead to their gait pattern significantly different. How to avoid all these confounding factors?

Author Response

We appreciate the time and efforts by the editor and reviewers in reviewing this manuscript. We have addressed all the issues indicated in the review report, and hopefully the revised version can meet the journal publication requirements

Reviewer 1 comments:

Overall: A full journal paper should cite approximately 40 relative references and minimal word count of 4000 words (please double check).

Introduction:

  1. The authors included background in the introduction section, but there is a lack of literature review in this study. The latest reference was cited in the introduction is 2020. There are many current studies focusing on DDH if you type the key words in Google scholar. The authors should essentially explain what the previous studies have done, and what their limitations were, and what the main contribution is in this study. Hence, the introduction section should be significantly improved, and should include more current references.

Thank you for your comment. Changes have been made on introduction and current references have been put in the introduction section.

I have a big concern about the study design and results.

Methods:

  1. How did you measure the speed, cadence, step width, step length and stride length, and how did you compare them?

Thank you for the comment. Measurement of speed cadence, step width, step length and strike length were measured with inertial motion capture system that calibrated when participant holding a neural pose (N-pose or T-pose). 3D motion analysis involves the analysis allowed speed. Cadence to be analyzed as part of basic gait parameters. Basic gait parameters were compared using mean±sd.

  1. Figure 1 is redundant. It repeated the contents in the first two paragraphs.

Thank you for the reviewer’s opinion on the matter of redundancy in Figure 1 and the text. As recommended by the International Council of Medical Journal Editors (ICMJE), particularly through the STROBE checklist that can be considered as the gold standard to follow for preparation of observational study manuscript, both detailed description of the numbers of individuals at each stage of study and using flow diagram to show the recruitment process are recommended. Thus, the authors prefer to abide by this gold standard and respectfully rebutted the reviewer’s anecdotal opinion.

  1. Figure 2 is not clear. Xsens Awinda should not have wires connecting sensors, but why the subject has wires secured?

Thank you for the comment. Corrections have been made accordingly at Figure 2.

  1. Given that the subjects have been recorded, why used the picture of Xsens from website, and not use the pictures of subject recruited in this study during their task performance?

Thank you for the comment. Due to confidentiality and the subject still under 18 which required subject parents or guardian permission, we are not allowed to use patients picture during their task performance.

  1. Stats analysis is not clear. T-test to analyze the mean values seems not correct. If you have two groups and each group has 2 subjects, the body weights are 120kg and 60kg in the first group, and 90kg and 90kg in the second group. The mean values are same, but the difference should be significantly different.

Thank you to the reviewer for sharing your perspective on using t-tests to analyze mean differences. The authors appreciate the reviewer's input and are open to discussing this further. The example that the reviewer provided, indeed described the concept of t-test analysis, where a t-value is calculated by comparing the difference between the means of the two groups while considering the variation within each group. However, the simplified scenario provided by the reviewer does not make sense because you just simply cannot compute any t-value without a mean difference. Furthermore, since the reviewer also never provided alternative statistical method in place of the t-test, the authors decided to respectfully disregard this comment entirely.

Results:

  1. For the stats analysis p=0.09, how did you get this result? There is ~10kg difference between these two groups. It is a big difference.

Thank you for the comment. Due to large standard deviations that can be seen in the weight for both groups, it suggests that the weight is more spread out which statistically, it will cause difference between the group is not significance.

  1. For figure 3, there are 3 curves in each plot. Are the 3 curves obtained from one subject or the average curves from all 14 subjects?

Thank you for the comment. The 3 curves are obtained from the average curves of all the 14 subjects.

  1. The description is not accurate for the statement of minimum hip abduction. For example, the minimum hip abduction for the red line is at 50 degrees. I don’t think there is a significant difference.

Thank you for the comment. Changes have been made accordingly in line 97-101 where authors have put the mean deviation to clearly highlight the significance difference between each groups in term of mean deviations.

  1. If the authors put knee and ankle results in the paper, you should describe them. But I did not see the description in the result section.

Thank you for the comment.  Changes have been made accordingly in line 104-106.

  1. I have a big concern about the subjects recruited in both groups. To compare the affected results and the control results for all the joints, there should not be a significant difference between the unaffected results and the control results between two groups. Otherwise, it will be difficult to identify the factors or obtain the conclusions for the comparison between the affected and control results. But from figure3, I can see some significant differences between the unaffected results and the control results. Moreover, the difference of anthropometric data between subjects may also lead to their gait pattern significantly different. How to avoid all these confounding factors?

Thank you for the comments. As unaffected limb will have compensatory affect due to the affected limbs (not statistically significant), to minute the compounding factor that maybe too minute for comparison, control groups was used. We are also aiming at a larger studies in future for standardizing the type of surgery for more detailed evaluation.

Round 2

Reviewer 2 Report

1. A full journal paper should cite approximately 40 relative references and minimal word count of 4000 words. I don't think this paper satisfies these basic requirements.

2. The introduction section was not significantly improved. Please follow the previous comments to expand the introduction.

Author Response

Reviewer 1 comments:

Overall: A full journal paper should cite approximately 40 relative references and minimal word count of 4000 words (please double check).

Introduction:

  1. The authors included background in the introduction section, but there is a lack of literature review in this study. The latest reference was cited in the introduction is 2020. There are many current studies focusing on DDH if you type the key words in Google scholar. The authors should essentially explain what the previous studies have done, and what their limitations were, and what the main contribution is in this study. Hence, the introduction section should be significantly improved, and should include more current references.

Thank you for your comment. Changes have been made to the introduction and current references have been put in the introduction section. Please refer to line 1 until 70. Changes also have been made accordingly regarding references number. Please refer to references heading. Thank You

I have a big concern about the study design and results.

Methods:

  1. How did you measure the speed, cadence, step width, step length and stride length, and how did you compare them?

Thank you for the comment. Measurement of speed cadence, step width, step length and strike length were measured with inertial motion capture system that calibrated when participant holding a neural pose (N-pose or T-pose). 3D motion analysis involves the analysis allowed speed. Cadence to be analyzed as part of basic gait parameters. Basic gait parameters were compared using mean±sd.

  1. Figure 1 is redundant. It repeated the contents in the first two paragraphs.

Thank you for the reviewer’s opinion on the matter of redundancy in Figure 1 and the text. As recommended by the International Council of Medical Journal Editors (ICMJE), particularly through the STROBE checklist that can be considered as the gold standard to follow for preparation of observational study manuscript, both detailed description of the numbers of individuals at each stage of study and using flow diagram to show the recruitment process are recommended. Thus, the authors prefer to abide by this gold standard and respectfully rebutted the reviewer’s anecdotal opinion.

  1. Figure 2 is not clear. Xsens Awinda should not have wires connecting sensors, but why the subject has wires secured?

Thank you for the comment. Corrections have been made accordingly at Figure 2.

  1. Given that the subjects have been recorded, why used the picture of Xsens from website, and not use the pictures of subject recruited in this study during their task performance?

Thank you for the comment. Due to confidentiality and the subject still under 18 which required subject parents or guardian permission, we are not allowed to use patients picture during their task performance.

  1. Stats analysis is not clear. T-test to analyze the mean values seems not correct. If you have two groups and each group has 2 subjects, the body weights are 120kg and 60kg in the first group, and 90kg and 90kg in the second group. The mean values are same, but the difference should be significantly different.

Thank you to the reviewer for sharing your perspective on using t-tests to analyze mean differences. The authors appreciate the reviewer's input and are open to discussing this further. The example that the reviewer provided, indeed described the concept of t-test analysis, where a t-value is calculated by comparing the difference between the means of the two groups while considering the variation within each group. However, the simplified scenario provided by the reviewer does not make sense because you just simply cannot compute any t-value without a mean difference. Furthermore, since the reviewer also never provided alternative statistical method in place of the t-test, the authors decided to respectfully disregard this comment entirely.

Results:

  1. For the stats analysis p=0.09, how did you get this result? There is ~10kg difference between these two groups. It is a big difference.

Thank you for the comment. Due to large standard deviations that can be seen in the weight for both groups, it suggests that the weight is more spread out which statistically, it will cause difference between the group is not significance.

  1. For figure 3, there are 3 curves in each plot. Are the 3 curves obtained from one subject or the average curves from all 14 subjects?

Thank you for the comment. The 3 curves are obtained from the average curves of all the 14 subjects.

  1. The description is not accurate for the statement of minimum hip abduction. For example, the minimum hip abduction for the red line is at 50 degrees. I don’t think there is a significant difference.

Thank you for the comment. Changes have been made accordingly in line 97-101 where authors have put the mean deviation to clearly highlight the significance difference between each groups in term of mean deviations.

  1. If the authors put knee and ankle results in the paper, you should describe them. But I did not see the description in the result section.

Thank you for the comment.  Changes have been made accordingly in line 104-106.

  1. I have a big concern about the subjects recruited in both groups. To compare the affected results and the control results for all the joints, there should not be a significant difference between the unaffected results and the control results between two groups. Otherwise, it will be difficult to identify the factors or obtain the conclusions for the comparison between the affected and control results. But from figure3, I can see some significant differences between the unaffected results and the control results. Moreover, the difference of anthropometric data between subjects may also lead to their gait pattern significantly different. How to avoid all these confounding factors?

Thank you for the comments. As unaffected limb will have compensatory affect due to the affected limbs (not statistically significant), to minute the compounding factor that maybe too minute for comparison, control groups was used. We are also aiming at a larger studies in future for standardizing the type of surgery for more detailed evaluation.
